# Matrix Metalloproteinases Contribute to the Calcification Phenotype in Pseudoxanthoma Elasticum

**DOI:** 10.3390/biom13040672

**Published:** 2023-04-12

**Authors:** Ricarda Plümers, Christopher Lindenkamp, Michel Robin Osterhage, Cornelius Knabbe, Doris Hendig

**Affiliations:** Herz- und Diabeteszentrum Nordrhein-Westfalen, Institut für Laboratoriums- und Transfusionsmedizin, Universitätsklinik der Ruhr-Universität Bochum, Georgstraße 11, 32545 Bad Oeynhausen, Germany

**Keywords:** Abcc6, calcification, matrix metalloproteinase, Marimastat, pseudoxanthoma elasticum

## Abstract

Ectopic calcification and dysregulated extracellular matrix remodeling are prominent hallmarks of the complex heterogenous pathobiochemistry of pseudoxanthoma elasticum (PXE). The disease arises from mutations in *ABCC6*, an ATP-binding cassette transporter expressed predominantly in the liver. Neither its substrate nor the mechanisms by which it contributes to PXE are completely understood. The fibroblasts isolated from PXE patients and *Abcc6*^−/−^ mice were subjected to RNA sequencing. A group of matrix metalloproteinases (MMPs) clustering on human chromosome 11q21-23, respectively, murine chromosome 9, was found to be overexpressed. A real-time quantitative polymerase chain reaction, enzyme-linked immunosorbent assay and immunofluorescent staining confirmed these findings. The induction of calcification by CaCl_2_ resulted in the elevated expression of selected MMPs. On this basis, the influence of the MMP inhibitor Marimastat (BB-2516) on calcification was assessed. PXE fibroblasts (PXEFs) exhibited a pro-calcification phenotype basally. PXEF and normal human dermal fibroblasts responded with calcium deposit accumulation and the induced expression of osteopontin to the addition of Marimastat to the calcifying medium. The raised MMP expression in PXEFs and during cultivation with calcium indicates a correlation of ECM remodeling and ectopic calcification in PXE pathobiochemistry. We assume that MMPs make elastic fibers accessible to controlled, potentially osteopontin-dependent calcium deposition under calcifying conditions.

## 1. Introduction

Patients in whom the rare multisystemic disorder pseudoxanthoma elasticum (PXE, OMIM 264800) has been diagnosed are mostly suffering from symptoms evoked by ectopic calcification, a hallmark of the disease. In most cases, the onset of the disease is characterized by the appearance of dermal yellow papular lesions and excessive wrinkling due to the calcification of elastic fibers [1,2]. Mineralization deposits especially in small vessels elicit atherosclerotic plaques in the cardiovascular system. Hypertension, *Claudication intermittent* and cardiomyopathy are further complications associated with the disease [3,4]. Calcification of the retinal Bruch’s membrane finally leads to fracturing, observed as angioid streaks, and the formation of hemorrhage in the ocular fundus. In the case of extreme damage to the retina, visual function may be restricted or lost [5,6].

It is assumed that a local imbalance between calcification inhibiting and promoting factors is the cause of the symptoms. Extracellular inorganic pyrophosphate (PP_i_) has been identified as restraining the local deposition of calcium and phosphate [7,8]. The hydrolytic activity of ectonucleotide phosphodiesterase 1 (ENPP1) leads to the conversion of ATP and, thus, an increasing pyrophosphate level in the extracellular space. In this way, ENPP1 represents another parameter in the calcification regulation network [9]. Patients with PXE feature a diminished plasma PP_i_ level [10,11,12]. Fibroblast cultures from PXE patients contain less cytosolic and extracellular PP_i_ and are characterized by a lower expression of ENPP1 [13,14]. The administration of PP_i_ prevents ectopic calcification in mice [12]. Bisphosphonates, which are stable metabolic PP_i_ analoga, are currently under investigation for therapeutic approaches in PXE [15,16]. The expression of osteopontin (OPN, also known as secreted phosphoprotein 1; gene name *SPP1*) is coupled with the presence of pyrophosphates [17,18]. OPN is a mineralization-associated protein for which an inhibitory effect on the growth of apatite crystals, consisting of calcium and phosphate, has been demonstrated [19,20].

The ectopic calcification is closely related to another hallmark of PXE: the defects in the extracellular matrix (ECM). Mineral apatite crystals in the core of elastic fibers were identified as disease-defining even before pathologic details or the genetic background were known [8,21,22,23]. Subsequently, several macromolecules were identified to colocalize to these deposits, including OPN [22]. Histologically, the calcification of the mid-dermal elastic fibers manifests in their fragmentation. Moreover, collagen fibers are deformed, and proteoglycans accumulate [24,25]. Matrix metalloproteinases (MMPs) are discussed to relate to the initiation of these ECM defects. The 23 MMP family members in humans and mice degrade a broad spectrum of ECM molecules, thereby contributing to a dynamic ECM remodeling [26,27]. An elevated serum concentration of MMP2 and MMP9 is measurable for PXE patients, while their fibroblasts exhibit a higher expression of MMP2 [28,29].

PXE is inherited autosomal-recessively [30]. Genetic defects in ATP-binding cassette transporter C6 (ABCC6) were associated with the disease for the first time in 2000 [31]. Since then, approximately 400 mutations resulting in the exhibition of PXE symptoms have been described [32]. Considering the systemic expression profile, ABCC6 is mainly expressed in the liver, more specifically in the basolateral plasma membrane of hepatocytes. Its expression in the affected peripheral tissues, such as the skin, is markedly lower [33]. Furthermore, the fibroblasts isolated from PXE patients show a disease-related phenotype even outside the systemic context. This includes a higher responsiveness to calcification stimuli, abnormal ECM remodeling, an altered metabolic lipid profile and cellular senescence [13,34,35,36]. A better understanding of the pathobiochemistry encompassing the whole organism was gained using an *Abcc6*^−/−^ mouse model. The model was generated by introducing a hygromycin resistance gene into exon 18 and disrupting the correct coding sequence of *mAbcc6*. These *Abcc6*^−/−^-mice resembled the clinical symptoms of PXE patients in large part [37].

This study aimed at a more detailed analysis of the deficits in human and murine fibroblasts with *ABCC6*-deficiency focusing on the extracellular matrix (ECM) and MMPs. Consequently, we analyzed ectopic calcification and ECM remodeling by treating human fibroblasts with the broad range MMP inhibitor Marimastat under calcifying conditions.

## 2. Materials and Methods

### 2.1. Human Cell Culture

Normal human dermal fibroblasts (NHDFs) were obtained from the Coriell Institute of Medical Research (Camden, NJ, USA). PXE fibroblasts (PXEFs), isolated from skin biopsies, were received as described previously [38]. The relevant information is summarized in Table 1.

Primary human dermal fibroblasts were cultured in Dulbecco’s modified Eagle’s medium (Thermo Fisher Scientific, Waltham, MA, USA) supplemented with 4 mM L-glutamine (PAN Biotech, Aidenbach, Germany), 1% (*v*/*v*) Penicilin-Streptomycin-Amphotericin B solution (100×; PAN Biotech, Aidenbach, Germany) and 10 % (*v*/*v*) fetal calf serum (Gibco, Waltham, MA, USA). The medium was changed twice per week. Meanwhile, fibroblasts were kept under standard conditions of 37 °C and 5 % CO_2_. Upon confluence, cells were subcultured using 0.05% (*v*/*v*) trypsin (PAN Biotech, Aidenbach, Germany).

In terms of the experimental procedure, human fibroblasts between passage eight and eleven were seeded at a density of 177 cells per mm^2^ and cultivated for 24 h in the complete medium supplemented with 10% fetal calf serum. The fibroblasts were cultured for a further 72 h prior to harvest for the basal expression analysis. According to the calcification experimental set-up, the exchange medium contained DMSO, 10 µM Marimastat (diluted in DMSO; Tocris, Bristol, UK), 8 mM CaCl_2_ or 10 µM Marimastat plus 8 mM CaCl_2_ for the calcification experiments. The concentration for the Marimastat application was determined in accordance with several publications [39,40,41]. The application of 8 mM CaCl_2_ for 21 days to trigger calcification was performed following our previously established protocol [42]. Studies on human fibroblasts were performed in biological and technical triplicates.

### 2.2. Murine Cell Culture

The *Abcc6*^−/−^ mouse model generated by Gorgels et al. [37], based on a hybrid model of C57BL/6 and 129/Ola, was backcrossed to C57BL/g for more than five generations. We used *Abcc6*^+/+^ C57BL/6 mice as the control animals. Housing, care and breeding were taken over by the central animal facility of Bielefeld University (Germany). The mice were kept with water and food ad libitum. In the present study, the animals were 4 to 6 weeks old. Fibroblasts were isolated from four male and four female animals for each genotype. The mice were anesthetized with isoflurane inhalation (1.5–2%) and killed by cervical dislocation. The animal preparation is in accordance with the law on animal welfare for scientific purposes in Germany because the animals were killed solely for tissue dissection without prior burden. Therefore, ethical approval of this part of the study was not required.

A 1 cm^2^ piece of skin was cut from the tail of the dead animals. The isolation of adult murine dermal fibroblasts was carried out according to Takashima et al. [43]. In brief, the skin was kept in a 4 mg/mL dispase in phosphate-buffered saline (PBS; Gibco, Waltham, MA, USA) for 4 to 6 h at room temperature. Following a washing step in PBS, the epidermis was deducted from the dermis. The latter was cut in 2 mm^2^ pieces and incubated in a 1 mg/mL collagenase solution (Sigma, St. Louis, MO, USA) in Dulbecco’s modified Eagle’s medium for 2 h at 37 °C. Subsequently, the reaction was stopped by adding the same amount of the complete murine fibroblast cultivation medium. The latter consisted of Dulbecco’s modified Eagle’s medium supplemented with 1 % (*v*/*v*) Penicillin-Streptomycin-Amphotericin, 4 mM L-glutamine, 20 % fetal calf serum, 1 % (*v*/*v*) non-essential amino acids (Gibco, Waltham, MA, USA) and 1 % (*v*/*v*) HEPES buffer (Gibco, Waltham, MA, USA). After filtering the cell suspension through a 100 µm strainer (Corning, Corning, NY, USA), cells were seeded at a density of 40 cells per mm^2^. The typical spindle-shaped morphology was confirmed via microscopy. Upon confluence, the murine fibroblasts were seeded for the experiment without further subculture at a density of 177 cells per mm^2^. Cells were harvested after 96 h. Studies on murine fibroblasts were performed in technical replicates.

### 2.3. Nucleic Acid Isolation

The RNA isolation from fibroblasts’ cultures was achieved using the NucleoSpin RNA Kit (Macherey-Nagel, Düren, Germany). The DNA for normalization was extracted utilizing the NucleoSpin Blood Extraction Kit (Macherey-Nagel, Düren, Germany). In both cases, the procedure was carried out in accordance with the manufacturer’s instructions. Nucleic acid concentrations were determined on a NanoDrop200 spectrophotometer (Peqlab, Erlangen, Germany).

### 2.4. Gene Expression Analysis

Quality control, RNA library preparation and sequencing on Illumina NovaSeq6000 for RNA sequencing were taken over by Novogene Co. (Beijing, China). The RNA from two cell lines per sex in biological triplicates was pooled in equal amounts for the analysis.

Regarding the real-time quantitative polymerase chain reaction (RT-qPCR), 1 µg of isolated RNA was initially transcribed into cDNA using SuperScript II Reverse Transcriptase (Thermo Fisher Scientific, Waltham, MA, USA). Measurements were performed with mixtures of 2.5 µL cDNA (1:10 diluted), 0.25 µL forward and reverse primers (Biomers, Ulm, Germany), 2.0 µL water and 5.0 µL LightCycler 480 SYBR Green I Master reaction mixture (Roche, Penzberg, Germany) in a LightCycler 480 system (Roche, Penzberg, Germany). Following an initial incubation for 5 min at 95 °C, the PCR program continued with 45 cycles of denaturation (95 °C, 10 s), annealing (specific annealing temperature, 15 s) and elongation (72 °C, 20 s). Finally, a melting curve analysis was conducted. The target-specific mRNA gene expression was relativized to the gene expression of a selection of three housekeeping genes. These were glyceraldehyde-3-phosphate-dehydrogenase (*hGAPDH*), hydroxymethylbilane synthase (*hHMBS*), ribosomal protein L13 (*hRPL13*) and succinate dehydrogenase complex flavoprotein subunit A (*hSDHA*) for studies in human cell lines. Eukaryotic translation initiation factor 3 subunit A (*mEif3a*), glyceraldehyde-3-phosphate dehydrogenase (*mGapdh*) and hypoxanthine phosphor-ribosyltransferase 1 (*mHprt*) gene expression were measured for relativization in studies in murine cell lines. The delta-delta Ct method considering PCR efficiency and internal calibration was applied. Studies on human fibroblasts were performed in biological and technical triplicates. Information and sequences of forward and reverse primers can be found in Table 2 and Table 3.

### 2.5. Enzyme-Linked Immunosorbent Assay

The protein concentrations of MMP3 and OPN in the cell culture supernatant were determined using commercially available ELISA Kits (DMP300 and DOST00; R&D Systems, Abingdon, UK). The results were normalized to the DNA content in the cell lysate.

### 2.6. hMMP12 Immunofluorescent Staining and Fluorescence Microscopy

Cells were seeded onto coverslips (Ø 13 mm) coated with 5 µg/cm^2^ rat collagen (ibidi, Gräfelfing Germany) for the fluorescence microscopy experiments. Following the cultivation procedure described above, the cells were washed twice in PBS, fixed for 20 min with 4% paraformaldehyde and washed twice again with PBS. Permeabilization was accomplished by 10 min incubation with 0.1% Triton^®^ X 100 (Roth, Karlsruhe, Germany) in PBS and another two washing steps. The blocking of unspecific binding sites was achieved by the application of 5% bovine serum albumin in PBS for 1 h. Cells were washed twice again with PBS. A primary antibody dilution in 1% bovine serum albumin in PBS was applied (ab137443, 1:100; Abcam, Cambridge, UK) for 2 h. Subsequently, another double washing step was completed before the cells were incubated for 1 h in the presence of a secondary antibody dilution in 1% BSA in PBS (ab150078, 1:400; Abcam, Cambridge, UK). After washing twice, the cells were counterstained with a 0.25 µM DAPI solution (Abcam, Cambridge, UK). After a final double wash step, the coverslips were mounted in ROTI^®^ Mount FluorCare mounting media (Roth, Karlsruhe, Germany). A BZ-X810 microscope (Keyence, Osaka, Japan) was used to capture fluorescence images.

### 2.7. Alizarin Red Staining

Cells were cultured in 24-well cell culture plates according to the experiment procedure stated above. The fibroblasts were washed twice with PBS and fixed by the addition of 70% ethanol and incubation for 30 s for staining. Subsequently, the cells were incubated for 5 min in a 40 mM alizarin red staining (ARS) solution (Sigma, St. Louis, MO, USA) and washed five times to remove the excess dye. The ARS and quantification were performed according to Gregory et al. [44]. In brief, the dye was dissolved out through incubation with 10% acetic acid for 30 min, and the contents of the well transferred into a microcentrifuge tube which was centrifuged at 18,000× *g* for 15 min. The vortexed mixture was then heated to 85 °C for 10 min and cooled down on ice for 5 min. Following another centrifugation step, 500 µL of the supernatant was neutralized with 200 µL 10 % (*v*/*v*) ammonium hydroxide. The absorbance of the mixture was measured in duplicates at 405 nm using a Tecan Reader infinite 200 Pro (Tecan, Männedorf, Switzerland).

### 2.8. ENPP1 Activity Assay

The ENPP1 activity was traced by a medium color change due to the conversion of its substrate thymidine 5′-monophosphate p-nitrophenyl ester. After fibroblasts were cultured, as stated above, the assay was executed as published by Lau et al. [45]. In summary, the cultivation medium was replaced by a medium containing 1 mg/mL substrate following a 1 h incubation at 37 °C. The absorbance of the cell culture supernatant at 415 nm was measured in duplicate using a Tecan Reader infinite 200 Pro (Tecan, Männedorf, Switzerland).

### 2.9. Statistical Analysis

The data of the mRNA gene expression analysis and protein concentration determination by ELISA were presented as mean ± standard error (SEM). The data concerning the ARS quantification and ENPP1 activity test were presented as box and whiskers with bars marking the 10–90 percentile. Statistical significance levels were calculated with non-parametric two-tailed Mann–Whitney U tests. Probability (*p*) values equal to or below 0.05 were assumed to be statistically significant.

## 3. Results

### 3.1. Upregulated Matrix Metalloproteinases Expression upon ABCC6-Deficiency

The degradation and fragmentation of ECM compounds in peripheral tissue are a hallmark of PXE. The grade and symptomatic manifestation of the resulting defects appear to be heterogenous between patients. Moreover, the exact underlying pathomechanism linking ECM irregularity to other hallmarks of this disease, such as the calcification, remains poorly understood.

Fibroblasts are the main cell type involved in the ECM maintenance of the affected peripheral tissue. Hence, we performed an RNA sequencing approach to obtain a better impression of intracellular dysregulations in female and male *ABCC6*-deficient fibroblasts. For this purpose, we used normal NHDFs as the healthy control as well as PXEFs. Moreover, fibroblasts isolated from a well-established mouse model [37] were utilized to further confirm our findings.

The RNA sequencing data revealed several gene ontology entities with differential expression patterns between NHDFs and PXEFs in female and male fibroblasts. Four of the twenty most significantly dysregulated terms in female PXEFs were directly associated with the ECM. In the case of the male human fibroblasts, this was true for half of the entities (Figure 1A). By further breaking down the list of differentially expressed genes, the group of MMPs stands out.

Fourteen MMPs were found to be expressed in significant amounts in human fibroblasts (fragments per kilobase of transcript per million fragments mapped (fpkm) > 0.1). *hMMP8*, *hMMP12*, *hMMP10*, *hMMP3*, *hMMP27*, *hMMP1* and *hMMP2* were overexpressed at least 1.5-fold in the male and female PXEFs compared to NDHFs. Except for *hMMP2*, the genes of these MMPs were located on a cluster at Chr11q21-24. Additionally, it should be mentioned that *hMMP24* and *hMMP15* RNA expressions were reduced by at least 50% in female PXEFs in relation to female NHDFs. *hMMP17*, *hMMP11*, *hMMP14*, *hMMP16* and *hMMP19* RNA expressions were neither diminished nor increased by more than 50% in male or female PXEFs compared to NHDFs (Figure 1B).

Eighteen MMPs were expressed in significant amounts (fpkm > 0.1) in the murine fibroblasts. *mMmp10*, *mMmp1a*, *mMmp13*, *mMmp1b* and *mMmp3* were overexpressed at least 1.5-fold in *Abcc6*^−/−^ fibroblasts in comparison to *Abcc6*^+/+^ fibroblasts. *mMmp8* and *mMmp17* expressions were upregulated exclusively in *Abcc6*^+/+^ fibroblasts derived from male animals, whereas the overexpression of *mMmp24* and *mMmp28* was observed only in female *Abcc6*^−/−^ fibroblasts. Surprisingly, *mMmp9* RNA expression was increased in male but diminished in female *Abcc6*^−/−^ fibroblasts relative to its expression in *Abcc6*^+/+^ fibroblasts. A difference in the expressions of *mMmp15*, *mMmp19*, *mMmp16*, *mMmp11*, *mMmp14*, *mMmp2*, *mMmp23* and *mMmp27* with more than 50% deviation was not detected when taking the Abcc6 genotype into account. Similar to the human gene cluster, the genes for *mMmp10*, *mMmp1a*, *mMmp13*, *mMmp1b*, *mMmp3*, *mMmp15* and *mMmp27* clustered on a locus on chromosome 9 (Figure 1C).

Subsequently, the mRNA expressional levels of the selected MMPs observed in the RNA sequencing data were validated via RT-qPCR. The selection was based on (a) the overall expression level, (b) the extent of differential expression between healthy and *ABCC6*-deficient fibroblasts and (c) the similarity in overexpression between female and male cell lines to summarize the results in a simplified way. Therefore, we defined the expressions of *hMMP1*, *hMMP3*, *hMMP10* and *hMMP12* for the human system and *hMmp1a*, *hMmp3* and *hMmp10* for the murine system as appropriate markers. hMMP3 and hMMP12 protein expression was determined using an ELISA or, respectively, immunofluorescent detection.

The comparison of the *hMMP1*, *hMMP3*, *hMMP10* and *hMMP12* mRNA expressions in the human cell lines all revealed a significant induction in PXEFs. It was increased by 5.1-fold (±0.9) for *hMMP1*, 10.5-fold (±2.4) for *hMMP3*, 3.9-fold (±0.6) for *hMMP10* and 6.6-fold (±1.5) for *hMMP12* (Figure 2A).

Differences in MMP mRNA expression patterns were not statistically significant in the murine cell lines but showed a similar tendency to those in human cells. *mMmp1a* mRNA expression was induced by 1.6-fold (±0.5), *mMmp3a* mRNA expression was elevated by 2.7-fold (±1.1) and *mMmp10* mRNA expression was enhanced by 9.5-fold (±3.7) in *Abcc6*^−/−^ fibroblasts towards *Abcc6*^+/+^ fibroblasts (Figure 2B).

An hMMP3-specific ELISA was utilized to confirm the overexpression on the transcriptional level in the human cell culture model. The hMMP3 protein concentration in cell culture supernatants normalized to the DNA content in the cell layer was found to be 0.4 ng/mL per µg_DNA_ (±0.1) for the NHDFs and 1.5 ng/mL per µgDNA (±0.4) for the PXEFs mirroring a 4.0-fold (±1.2) increase (Figure 2C).

The immunofluorescent labeling of hMMP12 demonstrates an overall higher load of hMMP12 in PXEFs compared to NHDFs, correlating similarly with *hMMP12* mRNA expression (Figure 2D). A localization in the endoplasmic reticulum is assumed due to the detection near the nucleus (white arrow) at a higher magnification (Figure 2E).

### 3.2. Correlation of the Calcification Phenotype and Matrix Metalloproteinases

Our results up to here confirm MMPs, a major regulator in ECM remodeling, as dysregulated upon ABCC6 deficiency. However, it remains unclear how this factor fits into the overall construction of PXE pathobiochemistry, namely, the ectopic calcification. Consequently, we applied a broad range MMP inhibitor, Marimastat, at a concentration of 10 µM alone or in combination with calcification culture conditions (21 days with 8 mM CaCl_2_) on human dermal fibroblasts.

Major changes in the cellular shape after 21 days were observed comparing the morphology of fibroblasts under control conditions, cultivation with 10 µM Marimastat or 8 mM CaCl_2_ or both in combination. NHDFs possessed a spindle-shaped cell body and grew to confluence under the control conditions. Partially overlapping cells made it difficult to recognize individual cells (blue arrow). Cultured in the control medium, PXEFs had flat cell bodies resulting in distinguishable cells and a lower cell density (red circle). These typical cell-shape phenotypes were maintained under cultivation with 10 µM Marimastat. When cultured under calcification-favorable circumstances, the morphologies of NHDFs and PXEFs became more similar. The cellular shape of NHDFs was marginally flattened, as seen in the cell in the blue circle exemplarily. Meanwhile, the PXEF morphology became more spindle-shaped and the cell density appeared to be higher (red arrow). An additional application of 10 µM Marimastat had no further influence on the cellular appearance in either NHDFs or PXEFs. Cultivation with 8 mM CaCl_2_ resulted in the deposits pointed out by white arrows (Figure 3).

In the following section, we study the fibroblasts treated in detail, beginning with the evaluation of MMP expression. The mRNA expression of *hMMP3*, *hMMP1* and *hMMP12* was measured and supported by the quantification of hMMP3 in the cell culture supernatant.

A comparison between the mRNA expressions of *hMMP3* in NHDFs and PXEFs revealed a 6.7-fold (±1.4) induction between the controls in a similar manner as described above and an 11.4-fold (±2.9) increase under calcifying conditions. While there was no significant change in the *hMMP3* mRNA expression upon the application of 10 µM Marimastat, the induction of calcification by 8 mM CaCl_2_ resulted in a 29.1-fold (±6.4) higher expression in NHDFs. A combined application of Marimastat and CaCl_2_ did not have any significant impact on this effect. Regarding PXEFs, there was similarly no alteration in the *hMMP3* mRNA expression under Marimastat treatment but an increase by 50.7-fold (±14.7) under calcifying conditions. In contrast to the NHDFs, the simultaneous administration of Marimastat and CaCl_2_ lowered the *hMMP3* mRNA expression significantly compared to a sole administration of calcium in PXEFs to 0.8-fold (±0.3) (Figure 4A).

The quantification of hMMP3 in the cell culture supernatants illustrates once more a significantly higher expression in PXEFs relative to that in NHDFs under the control (4.7-fold ± 1.7) and calcifying (39.3-fold ± 18.5) conditions. The amount of hMMP3 was not significantly altered under the application of 10 µM Marimastat in either NHDFs or PXEFs but a higher load of hMMP3 was recognized for both when culturing the fibroblasts with 8 mM CaCl_2_, namely 10.8-fold (±3.5) in NHDFs and 89.6-fold (±45.0) in PXEFs. A combined application of Marimastat and calcium exhibited no significant reduced hMMP3 expression in either NHDFs or PXEFs (Figure 4B).

The gene expression analysis of *hMMP1* disclosed once again a significant elevation by 4.1-fold (±0.6) in PXEFs towards NHDF that was not mirrored under calcification. A gain in *hMMP1* gene expression was determined under treatment with 8 mM CaCl_2_ in NHDFs (14.4-fold ± 2.7) and PXEFs (4.3-fold ± 0.8). By contrast, the application of Marimastat under non-calcifying conditions resulted in a modified *hMMP1* expression neither in NHDF nor in PXEF. Under calcifying conditions, supplementation with Marimastat significantly reduced the *hMMP1* expression in NHDFs to 0.1-fold (±0.1) and in PXEFs to 0.7-fold (±0.1) (Figure 4C).

As a third marker, *hMMP12* gene expression was considered. The NHDFs and PXEFs differed in their *hMMP12* expression, reflected by a 28.8-fold (±7.0) and 14.7-fold (±4.1) induction under calcifying conditions, respectively. While a treatment with 10 µM Marimastat had no influence on this factor in either of the cell types, the application of 8 mM CaCl_2_ resulted in NHDF and in PXEFs in a 12.0-fold (±2.1) and a 6.1-fold (±2.0) higher *hMMP12* mRNA level, respectively. The PXEFs simultaneously treated with Marimastat and calcium did not feature a modified *hMMP12* gene expression, but the same conditions lead to a significant decline of *hMMP12* RNA in NHDFs (0.18-fold ± 0.04) (Figure 4D).

Having evaluated the course of MMP expression in our experimental set-up, our next step was to assess the influence of the MMP inhibitor Marimastat administration on calcification. Therefore, we performed a quantitative ARS, determined the gene expression of *hSPP1* and *hENPP1* and confirmed the results with an hOPN ELISA and an ENPP1 activity assay.

Alizarin red, a dye to color calcium deposits, was used to evaluate the grade of calcification in the fibroblast monolayers. Exemplary pictures demonstrate the successful staining and a stronger redness of cultures treated with CaCl_2_ (Figure 5A).

A subsequent quantification of the staining attested this impression. Both NHDFs and PXEFs exhibited a higher grade of staining after cultivation with 8 mM CaCl_2_ compared to cultivation under control conditions: the staining was 2.4-fold (±0.6) stronger in NHDFs and 20.8-fold (±5.2) stronger in PXEFs. The basal calcification mapped by the degree of dyeing was found to be significantly lower in PXEF than in NHDF (0.5-fold ± 0.1) but higher under calcification conditions (4.6-fold ± 1.3). The additional administration of 10 µM Marimastat had no significant influence on the staining in non-calcifying circumstances. Marimastat and calcium in combination triggered a gain in the dyeing in comparison to cultivation with CaCl_2_ alone. For NHDF, the factor for this variance amounted to 1.8-fold (±0.4), and for PXEF, 1.9-fold (±0.5) (Figure 5B).

The relative *hSPP1* gene expression, a marker for the mineralization of tissues, did not differ significantly under control conditions between NHDFs and PXEFs. The additional application of 10 µM Marimastat had no influence on the control status. By contrast, the *hSPP1* mRNA expression was significantly enhanced when cultivation was carried out in the presence of 8 mM CaCl_2_ in both NHDFs (6.4-fold ± 1.1) and PXEFs (1.8-fold ± 0.4). By direct comparison, PXEFs exhibited a 0.5-fold (±0.1) reduced *hSPP1* gene expression within this condition. Beyond this observation, a further addition of Marimastat enlarged the *hSPP1* gene expression increase by another 2.3-fold (±0.8) in NHDFs and 1.7-fold (±0.3) in PXEFs compared to CaCl_2_ administration only (Figure 6A).

An hOPN ELISA was conducted with cell culture supernatant samples to substantiate the gene expression results. A significant difference in the hOPN concentration in the supernatants between NHDFs and PXEFs was not ascertainable in direct comparison under control and calcifying conditions or in the presence of 10 µM Marimastat. Nevertheless, culturing the fibroblasts with 8 mM CaCl_2_ provoked a rise in the hOPN concentration by 53.7-fold (±23.7) in NHDFs and 49.7-fold (±22.3) in PXEFs. A combined treatment with Marimastat and calcium further enhanced, although not significantly, the accumulation of hOPN in the supernatants of NHDFs and PXEFs by 2.9-fold (±1.6) and 2.6-fold (±1.2), respectively (Figure 6B).

Regarding the *hENPP1* gene expression, a representative marker for calcification inhibition and regulation, a significantly diminished value was determined in PXEFs compared to NHDFs under the control (0.6-fold ± 0.1) and calcifying conditions (0.2-fold ± 0.1). The cultivation only with 10 µM Marimastat had no additional effect at that point in either of the fibroblast types. While the administration of 8 mM CaCl_2_ did not influence the *hENPP1* mRNA level in PXEFs, the calcifying condition resulted in its 1.6-fold (±0.2) increase in NHDFs. Furthermore, Marimastat and CaCl_2_ in combination evoked an induction of *hENPP1* mRNA expression by 1.4-fold (±0.2) in NHDFs and 1.3-fold (±0.2) in PXEFs (Figure 6C).

An ENPP1 activity assay, based on the implementation of its substrate thymidine 5′-monophosphate p-nitrophenyl ester, was performed to strengthen the gene expression results. Both NHDFs and PXEFs were found to have an equal ENPP1 activity in control cultures that was not influenced by 10 µM Marimastat. NHDF exhibited a slight, yet not significant, rise in ENPP1 activity by 1.2-fold (±0.1) in a calcifying surrounding brought about by 8 mM CaCl_2_, while this parameter was reduced significantly by 0.6-fold (±0.1) in PXEFs under the same condition. Supplementation with 10 µM Marimastat had no impact on the ENPP1 activity in PXEFs treated with CaCl_2_ but led to an even higher ENPP1 activity in NHDFs (1.8-fold ± 0.3) compared to only CaCl_2_ cultivation (Figure 6D).

## 4. Discussion

The biggest challenge in adequately managing PXE still remains the understanding of its basic pathomechanisms. As the disease is characterized by a complex multisystemic pathobiochemistry influenced by a variety of genetic modifiers [46] not only the identification of the ABCC6 substrate, assumed to be predominantly transported into the bloodstream by hepatocytes [47], is of the utmost interest. Instead, the research on and connection of all resulting related pathomechanisms will enable us to ensure good care for the patients. These are as diverse as the manifestation of the disease itself and include the ectopic calcification in the dermis, retinal and small vessels [48,49]; the abnormal remodeling of the ECM [24,25,50]; altered lipid metabolisms [35,51,52]; premature aging and senescence [53,54]; and a proinflammatory phenotype [36]. The present study focused on a better understanding between the ectopic calcification and the abnormal dermal ECM remodeling.

Our first approach was to identify predominantly altered gene ontologies in human and murine dermal fibroblasts with ABCC6 deficiency by RNA sequencing to assess reliable markers for a comparison. As expected, due to the phenotype of these exhibited fibroblasts described previously [29,50], modifications in ECM-related ontologies were prevailing. A profound examination of the genes affected revealed striking changes in the MMP expression. The involvement of MMPs in PXE pathobiochemistry was unsurprising. MMP2 has been outlined to raise the degradative potential of human PXEFs [29]. Patients feature elevated MMP2 and MMP9 serum levels, and an MMP2 promoter polymorphism is suspected as a genetic modifier for the disease [28,55].

However, to date, no extensive study on the expression of the whole MMP family has been carried out. RNA sequencing allowed us to close this gap. Our data confirmed a slight mRNA overexpression of MMP2 in humans, but not in murine fibroblasts. Instead, we found the gene expression of *hMMP8*, *hMMP12*, *hMMP10*, *hMMP3*, *hMMP27* and *hMMP1* to be increased. The fact that these genes clustered on chromosome 11q21-23 was remarkable [56]. An RNA sequencing analysis of fibroblasts isolated from the murine *Abcc6*^−/−^ mouse model further strengthened this observation. Here, the expressions of *mMmp10*, *mMmp1a*, *mMmp13*, *mMmp1b* and *mMmp3* were elevated compared to fibroblasts isolated from wild-type mice. Together with *mMmp15* and *mMmp27,* these genes clustered on chromosome 9. Here, the close orthologous relation between the human and the murine gene clusters as well as the corresponding MMPs had to be distinguished. The cellular function of the individual MMPs in these groups was negligible due to their large substrate diversity [26]. Furthermore, minor differences in the expression patterns within these clusters might be due to species-specific regulation and display a point for further investigation. Affirmative representative measurements of the *mMmp1a*, *mMmp3* and *mMmp10* mRNA gene expression in murine fibroblasts via RT-qPCR went along with the results from RNA sequencing, although they were not statistically significant. This fact might be due to the high variances between the murine fibroblast cultures. The results gained from the murine cell cultures should be considered as additional supportive data in the present study as we were limited in the number of murine cell lines. As this was a limitation of our study, further studies focusing on the role of MMPs in the murine model are of high interest.

Therefore, we focused on the human fibroblast cultures in the remaining parts of the study. For these, we successfully demonstrated a significant increase in mRNA expression levels of *hMMP1*, *hMMP3*, *hMMP10* and *hMMP12* by a multiple. The findings could be transferred to the protein level, demonstrated by the quantification of MMP3 in the cell culture supernatant and immunofluorescent staining of MMP12. Consequently, a strong correlation between MMP mRNA expression and translation into protein was indicated. Moreover, the nuclear near-detection of MMP12 might correspond to localization in the endoplasmic reticulum and subsequent secretion to the extracellular space. This fits with the characteristic that MMPs are predominantly secreted except from individual membrane-bound family members (such as MMP14 and MMP15) [57]. Although MMPs are regulated at several stages, including post-translational secretion and activation, the transcriptional regulation is of the greatest importance, as our results regarding PXEFs confirm [58].

It is worthwhile to take a common expression regulation into account for the finding that a cluster of MMPs is overexpressed due to ABCC6 deficiency. The 5′ promotors’ regions of the MMP genes involved share a TATA Box and, except for *hMMP8*, binding sites for activator protein 1 (AP-1) and polyoma enhancer A-binding protein 3 (PEA3) [58]. The binding of the two corresponding transcription factors is a necessary factor in MMP gene transcription modulated by exogenous signals, such as cytokines, growth factors or cell matrix interactions [59,60]. The factor decisive for the overexpression in the present case can only be assumed and requires further studies. A potential mediator might be interleukin-1β, a proinflammatory cytokine, which is known to induce MMP expression and is associated with a severe course of PXE [61,62]. Interestingly, this induction process was observed in calcific aortic valve stenosis, an age-related simultaneous occurrence of calcification and ECM remodeling [63]. However, the binding of activator elements in the promotor region might not be the only factor contributing to the gene expression regulation of MMPs. The hypomethylation of the promotor region resulted in the overexpression of several MMPs. This was notably proven for human osteoarthritic chondrocytes: another disease featuring abnormal ECM dynamics and calcification [64]. Taking these considerations into account, a calcification-dependent induction of cluster Chr11q21-23 MMPs’ expression by a calcification response element in the promotors is conceivable.

Against this background, the fact that we observed a potent induction of exemplarily measured *hMMP1*, *hMMP3* and *hMMP12* mRNA expression and hMMP3 content in cell culture supernatants for fibroblasts cultured with CaCl_2_ fits well. The induction was detectable for NHDFs and PXEFs, although the final amount was higher in PXEFs due to the increased basal expression. A general association of, especially, vascular calcification and MMPs has been described primarily for rodents. Subsequently to the induction of aortic calcification in rats, the administration of the MMP inhibitor GM6001 resulted in the reduced calcification of arteries [65]. Another rat model, in this case displaying vascular calcification in chronic kidney diseases, exhibited a raised MMP-2 and MMP-9 expression and activity upon arterial calcification [66]. The MMP knock-out mice did not respond to the administration of CaCl_2_ with arterial calcification [67]. The hypothesis which emerged from these studies was, therefore, that the MMPs made elastic fibers of arteries accessible to calcium deposits, thereby linking these two hallmarks of PXE pathobiochemistry. Moreover, the linkage was underlined by an approach to intervene in the DNA damage response of PXEFs, resulting in the reduced *hMMP2* expression and diminished calcification simultaneously [68].

We created an in vitro cultivation set-up to evaluate the status of human fibroblasts treated with Marimastat under standard and calcifying conditions to apply the assumption of a correlation between calcification and MMP-based elastic fiber degradation to our model. Marimastat or BB2516 is a broad range MMP inhibitor initially developed to treat migration-dependent metastasis in cancer [69]. It should be noted that the targeted inhibition of individual MMPs could be a focus for further studies to specify their functional role in the biochemistry of PXE.

Here, the application of Marimastat led to marginal changes but no full repression in MMP expression, which goes along with the fact that Marimastat inhibits the MMP activity by mimicking the substrate and competing with it rather than intervening in its expression [70]. No differences in morphology in either NHDFs or PXEFs were observed upon Marimastat treatment. The administration of the calcification medium had two effects on the appearance properties of the cultures. Firstly, massive deposits, later identified as calcium-containing by ARS, were observed in the NDHF and PXEF cultures. Secondly, PXEFs showed a decline in their senescence-associated morphology represented by the spread shape turning into a spindle shape. The opposite was observed for NHDFs. A possible conclusion might be that PXEFs are adapted to calcification conditions, which is reflected by their higher responsiveness to calcification stimuli [13]. Future studies considering the proliferation and migration rates of NHDF and PXEF under calcifying conditions may provide further insight into the general adaptation of fibroblasts to these circumstances.

We assessed the calcification status of the fibroblasts by the quantification of alizarin red-stained deposits; the expression of the apatite crystal formation-inhibitor OPN; and the mRNA expression and activity of ENPP1, which are relevant in the PP_i_ household and, thereby, prevent calcification. As expected, calcifying conditions induced calcium deposition, as well as regulating mechanisms of calcification displayed by OPN expression and, in the case of NHDF, ENPP1 expression and activity. Upon cultivation with calcium, the higher potential for PXEFs to calcify was displayed by a stronger ARS, a lower mRNA expression of *hSPP1* and *hENPP1* and a reduced ENPP1 activity compared to NHDFs. These data resemble the observations of failing calcification restriction by inhibitors in PXEFs made previously [13,14].

The ambivalent results gained under combined treatment with Marimastat were of greater interest.

The application of the MMP inhibitor provoked an extended deposition of calcium in both NHDFs and PXEFs. This observation was rather surprising as it was stated that MMP inhibitors can prevent calcification in vivo [65,66]. In addition, it must be considered that MMPs do not only affect elastic fibers, even though these are essentially affected by ECM calcification in PXE. The overexpression of MMPs reduces cell–matrix contact. This has been shown to induce the calcification of vessel walls by vascular smooth muscle cells [71,72]. In further experiments, this relationship could be elucidated by culturing on culture surfaces coated with different ECM components.

Furthermore, the Marimastat treatment caused an induction of OPN expression in NHDFs and PXEFs, and thereby contributed to the prevention of apatite crystal growth. Moreover, it led to a gain in ENPP1 gene expression and activity in NHDFs, similarly supporting an anti-calcifying effect. It should be noted that PXEFs did not respond to Marimastat or the CaCl_2_ stimuli by adjusting their PP_i_ household through ENPP1. The latter is compliant with the declining PP_i_ availability not only in fibroblasts’ cultures but also in the serum of PXE patients [11,13,14].

The fact that *hSPP1* has been identified as a modifier gene in PXE reflects that OPN is critical for limiting the calcification process in this disease [73]. A hypothetical explanation for our surprising findings that MMP inhibition enhances calcification, and MMPs might as well have a regulatory effect on ectopic calcification, might therefore be found in the association of OPN and elastic fibers. A fine analysis of the calcified and fragmented elastic fibers performed by Contri et al. revealed an association of OPN and other calcium-binding proteins to the mineral deposits [22]. In the case of a calcium overflow, the fibroblasts may react with the secretion of MMPs and, thus, make elastic fibers accessible for calcium association with proteins, such as OPN. This hypothesis is strengthened by a study using aluminum to pre-treat elastic fibers and, thereby, prevent the degradation by MMPs and calcium deposition in the fiber core [67]. Apart from that, the consequence might be that no excess calcium is intercepted by incorporation into the elastic fibers, where OPN restricts further crystal growth. In this way, the calcification of elastic fibers would represent a mechanism to handle calcium overload by fibroblasts.

The inhibition of elastic fiber-based calcium interception in our cell culture model may have led to the strengthening of other anti-calcifying processes, such as ENPP1 expression. Nonetheless, since calcification is based on an enormous regulatory network [74,75], it cannot be ruled out that other mechanisms not considered in this work will come into play in this case.

The fact that an in vitro model is limited when mirroring the in vivo situation should be considered. Future research may focus on the ultrastructure of elastic fibers under MMP inhibitor treatment and the systemic consequences.

In conclusion, our study gives detailed insight into the contribution of MMPs to calcification in the pathomechanism of PXE. Not only was a whole spectrum of MMP genes belonging to the human cluster Chr11q21-23 overexpressed in PXEFs, but those MMPs were also strongly associated with the induction of calcification. The results were strengthened by the overexpression of a related MMP gene cluster on chromosome 9 in the *Abcc6***^−/−^** mouse model. The treatment with Marimastat confirmed the synergy between matrix degradation and calcification. Contrary to expectations, Marimastat did not abolish excessive calcification but reinforced it, giving a first hint regarding a defanging of excess calcium by the incorporation and binding to OPN inside elastic fibers.

## Figures and Tables

**Figure 1 biomolecules-13-00672-f001:**
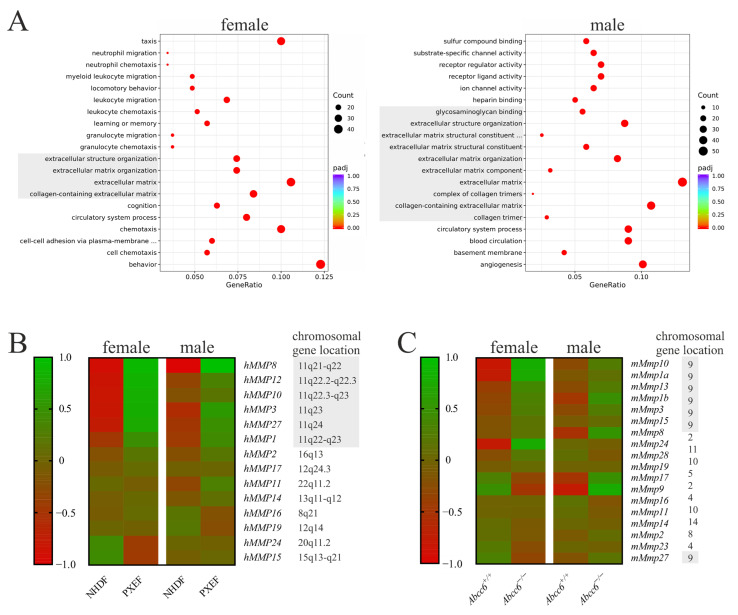
RNA sequencing data from human and murine fibroblasts with or without ABCC6 deficiency. (**A**) Differential expression pattern according to the gene ontology database in PXEFs compared to NHDFs assessed separately for female (NHDF n = 2, PXEF n = 2) and male cell lines (NHDF n = 2, PXEF n = 2). Entities directly associated with the extracellular matrix are greyed. (**B**) Heat map of the differential expression in female and male human NHDFs and PXEFs regarding MMPs plus the specification of the respective gen locus. A location on the gene cluster Chr11q21-24 is highlighted in grey. (**C**) Heat map of differential expression in murine *Abcc6*^+/+^ and *Abcc6*^−/−^ fibroblasts from female (*Abcc6*^+/+^ n = 4, *Abcc6^−/−^* n = 4) and male (*Abcc6*^+/+^ n = 4, *Abcc6*^−/−^ n = 4) animals regarding MMPs plus the specification of the respective gen locus. A location on the gene cluster Chr9 is highlighted in grey.

**Figure 2 biomolecules-13-00672-f002:**
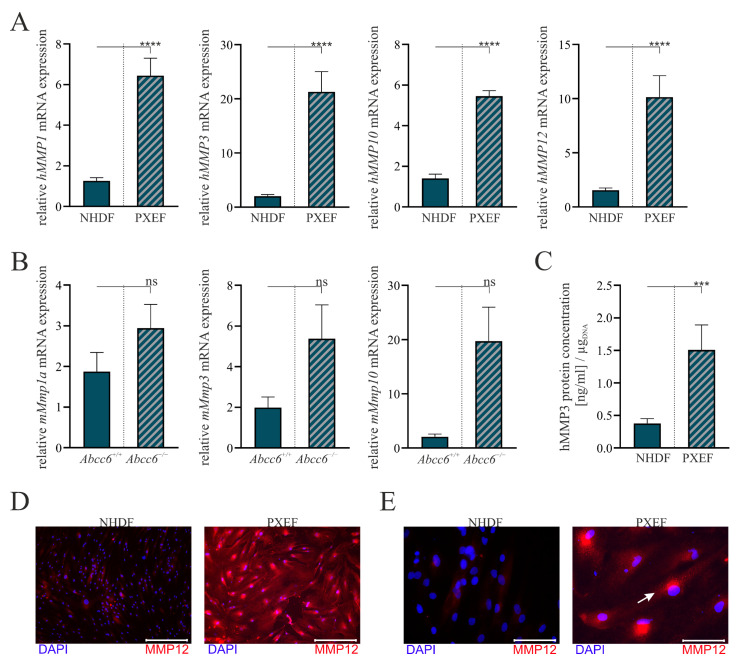
The MMP expression in human and murine fibroblasts with or without ABCC6 deficiency. Fibroblasts were cultured for 72 h in delipidated medium prior to the harvest. (**A**) mRNA expression of *hMMP1*, *hMMP3*, *hMMP10* and *hMMP12* in NHDFs (n = 4) and PXEFs (n = 4) and (**B**) mRNA expression of *mMmp1a*, *mMmp3* and *mMmp10* in fibroblasts derived from *Abcc6*^+/+^ or *Abcc6*^−/−^ mice were measured via RT-qPCR. (**C**) hMMP3 protein concentration in the cell culture supernatant of NHDFs (n = 4) and PXEFs (n = 4) was evaluated utilizing an ELISA and normalizing it to the DNA content in the cell layer. (**D**,**E**) Immunofluorescent staining of hMMP12 (red) and nuclear counterstaining with DAPI (blue) of NHDF and PXEF monolayers. Representative images are shown in 10× magnification ((**D**), scale bar 400 µm) and 40× magnification ((**E**), scale bar 100 µm). Data are shown as mean ± SEM. Mann–Whitney U test significance levels: not significant (ns), *p* < 0.001 (***), *p* < 0.0001 (****).

**Figure 3 biomolecules-13-00672-f003:**
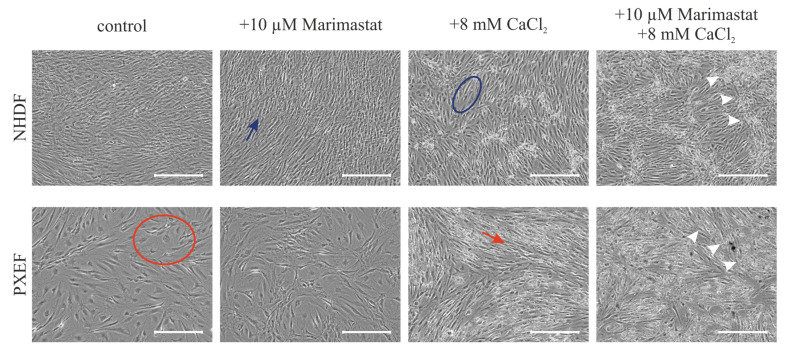
Representative bright-field microscopy pictures of NHDFs and PXEFs cultured for 21 days. Cultivation conditions included control treatment with DMSO, 10 µM Marimastat, 8 mM CaCl_2_ or a combination of 10 µM Marimastat and 8 mM CaCl_2_. Exemplary calcium deposits are highlighted with white arrows. The blue arrow points to an exemplary healthy fibroblast spindle-shape while the blue circle marks an exemplary healthy fibroblast with extended cytoplasm. The red circle surrounds an exemplary PXEF with a flat cell body whereas the red arrow points to a PXEF with an acquired spindle-shape. Scale bars: 400 µm.

**Figure 4 biomolecules-13-00672-f004:**
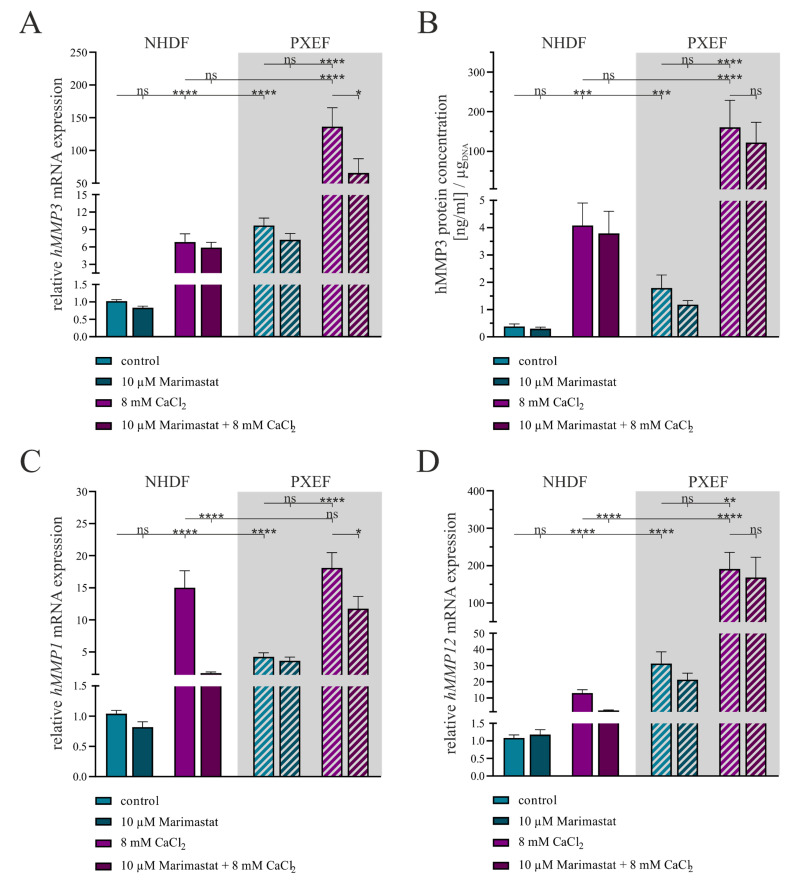
The expression of MMPs in human dermal fibroblasts under separate or combined treatment with 10 µM Marimastat and 8 mM CaCl_2_ for 21 days. The NHDFs (n = 3) and PXEFs (n = 3, grey highlighted) were kept under the conditions specified for 21 days. (**A**) mRNA expressions of *hMMP3*, (**C**) *hMMP1* and (**D**) *hMMP12* were measured via RT-qPCR. (**B**) hMMP3 protein concentration in the cell culture supernatant of NHDFs (n = 3) and PXEFs (n = 3) was evaluated utilizing an ELISA and normalizing it to the DNA content in the cell layer. Data are shown as mean ± SEM. Mann–Whitney U test significance levels: not significant (ns), *p* < 0.05 (*), *p* < 0.01 (**), *p* < 0.001 (***), *p* < 0.0001 (****).

**Figure 5 biomolecules-13-00672-f005:**
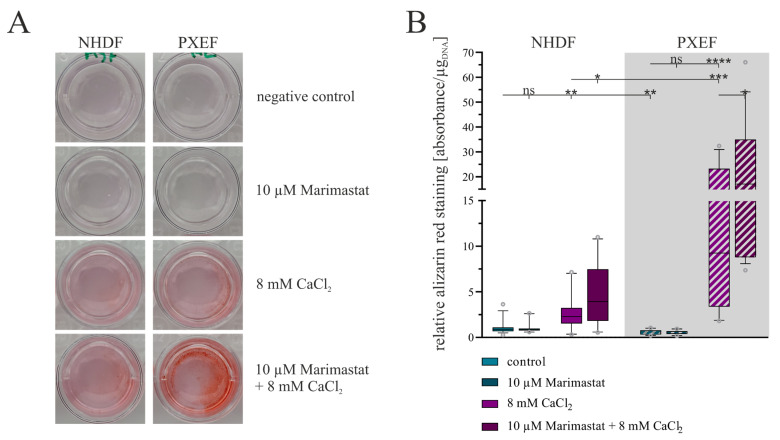
Evaluation of the calcification status in human fibroblasts concerning separate or combined treatment with 10 µM Marimastat and 8 mM CaCl_2_ for 21 days. The NHDFs (n = 3) and PXEFs (n = 3, grey highlighted) were kept under the conditions specified for 21 days. (**A**) Exemplary pictures of fibroblasts’ cell culture wells after ARS and (**B**) its quantification normalized to the DNA content in the cell layer and relative to the staining in control NHDFs. Data are shown as mean and boxes marking the 10–90 percentile. Mann–Whitney U test significance levels: not significant (ns), *p* < 0.05 (*), *p* < 0.001 (**), *p* < 0.001 (***), *p* < 0.0001 (****).

**Figure 6 biomolecules-13-00672-f006:**
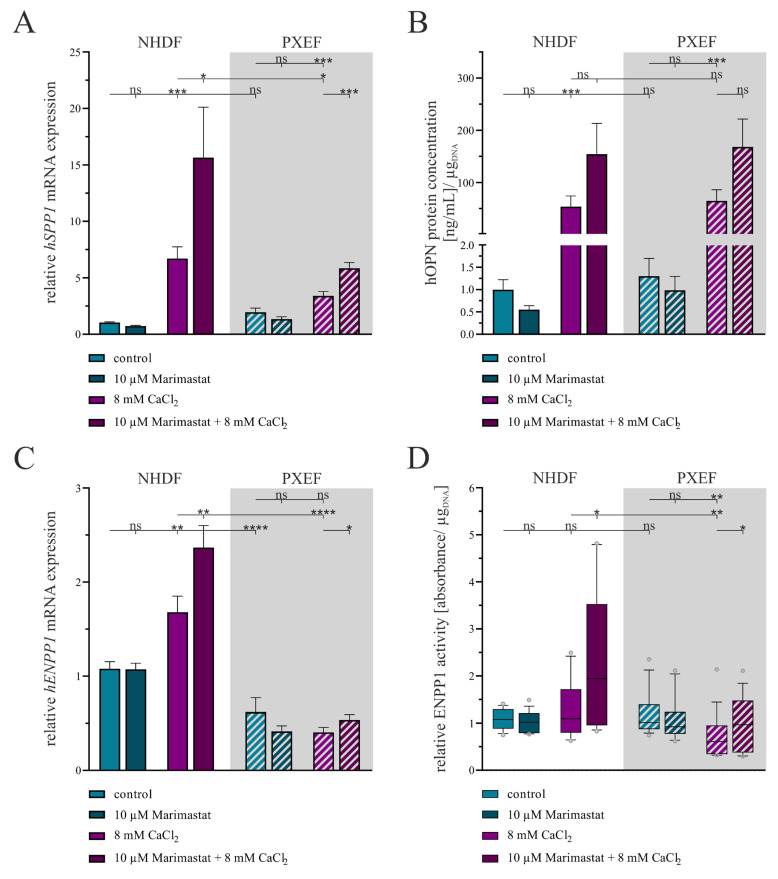
Evaluation of the calcification status in human fibroblasts concerning separate or combined treatment with 10 µM Marimastat and 8 mM CaCl_2_ for 21 days. The NHDFs (n = 3) and PXEFs (n = 3, grey highlighted) were kept under the conditions specified for 21 days. (**A**) Relative *hSPP1* mRNA gene expression measured by RT-qPCR and (**B**) hOPN concentration in cell culture supernatant determined by ELISA and normalized to the DNA content in the cell layer. (**C**) Relative *hENPP1* mRNA expression assessed by RT-qPCR and (**D**) ENPP1 activity in the cell culture supernatant normalized to the DNA content in the cell layer. (**A**–**C**) Data are shown as mean ± SEM or (**D**) as mean and boxes marking the 10–90 percentile. Mann–Whitney U test significance levels: not significant (ns), *p* < 0.05 (*), *p* < 0.001 (**), *p* < 0.001 (***), *p* < 0.0001 (****).

**Table 1 biomolecules-13-00672-t001:** Characteristics of fibroblasts used in this study. SSM, splice site mutation; hm, homozygous; cht, compound heterozygous; wt, wild type; n.s., not specified.

Sample ID	Gender	Age [year]	Biopsy Source	ABCC6 Genotype		Genotype Status
PXEF						
P265F	female	62	neck	c.1132C>T (p.Q378X)	c.3421C>T (p.R1141X)	cht
P3M	male	57	neck	c.3421C>T (p.R1141X)	c.3883-6G>A (SSM)	cht
P128M	male	51	neck	c.3769_3770insC (p.L1259fsX1277)	c.3769_3770insC (p.L1259fsX1277)	hm
P255F	female	48	arm	c.3421C>T (p.R1141X)	c.2787+1G>T	cht
P308M	male	42	n. s.	c.3421C>T (p.R1141X)	c.-90ins14	cht
**NHDF**						
F63A (AG12786)	female	63	arm	-	-	wt
M57A (AG13145)	male	57	arm	-	-	wt
M52A (AG11482)	male	52	arm	-	-	wt
F48A (AG14284)	female	48	arm	-	-	wt
M42A (AG06307)	male	42	arm	-	-	wt

**Table 2 biomolecules-13-00672-t002:** Sequences, annealing temperatures (TA), efficiencies and resulting product sizes of primers used for qRT-PCR in studies based on human fibroblasts.

Gene	5′-3′ Primer Sequences	TA (°C)	Efficiency	Product Size (bp)
*hENPP1*	AATGCCCCTTTGGACATC CCCGTAACTCACTTTGGT	59	1.72	151
*hGAPDH*	AGGTCGGAGTCAACGGAT TCCTGGAAGATGGTGATG	63	1.84	223
*hHMBS*	CTGCCAGAGAAGAGTGTG AGCTGTTGCCAGGATGAT	63	1.92	165
*hMMP1*	AGAAACACAAGAGCAAGATGTG TGGCGTGTAATTTTCAATCCTGT	63	1.85	298
*hMMP3*	GCCATCTCTTCCTTCAGGCG CCTAGGGTGTGGATGCCTCT	63	1.85	141
*hMMP10*	GCATTCAGTCTCTCTACGGACC AAAAACGGTGTCCCTGCTGT	66	2.00	287
*hMMP12*	CACATTCAGGAGGCACAAAC ATTTCCCACGGTAGTGACAG	59	1.82	275
*hRPL13*	CGGAAGGTGGTGGTCGTA CTCGGGAAGGGTTGGTGT	63	1.87	115
*hSDHA*	AACTCGCTCTTGGACCTG GAGTCGCAGTTCCGATGT	63	1.93	177
*hSPP1*	TGATGACCATGTGGACAG ACCATTCAACTCCTCGCT	59	1.70	322

**Table 3 biomolecules-13-00672-t003:** Sequences, annealing temperatures (TA), efficiencies and resulting product sizes of primers used for qRT-PCR in studies based on murine fibroblasts.

Gene	5′-3′ Primer Sequences	TA (°C)	Efficiency	Product Size (bp)
*mEif3a*	GAGTATCAGGAGCGAGTCAAG CCTCTCATCATCCCGAGTTTC	59	1.72	151
*mGapdh*	GCATCTTGGGCTACACTGAGG GGGTGGTCCAGGGTTTCTTAC	59	1.92	211
*mHprt*	GTTGGATACAGGCCAGAC GCCACAGGACTAGAACAC	59	1.93	226
*mMmp1a*	GTTGGAGCAGGCAGGAAGGAGG ACCCCACACCTGGGCTTCTT	68	2.00	328
*mMmp3*	CCCCTGATGTCCTCGTGGTA GTGCCCTCGTATAGCCCAGA	66	2.00	150
*mMmp10*	GTACCTTCCCAGGTTCGCCA GGGTAAAAGTCTCCGTGTTCTCC	66	2.00	217

## Data Availability

The original raw data and materials presented in the study will be made available upon request. Further inquiries can be directed to the corresponding author.

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
