# Peer review of "Matrix Metalloproteinases Contribute to the Calcification Phenotype in Pseudoxanthoma Elasticum"

_biomolecules, 2023, doi:10.3390/biom13040672_

Round 1

Reviewer 1 Report

The manuscript by Plümers et al entitled “Matrix metalloproteinases contribute to the calcification phenotype in Pseudoxanthoma elasticum” shows studies and experiments performed focus on understanding the ectopic calcification and the abnormal dermal ECM remodeling in Pseudoxanthoma elasticum using NHDFs (Normal human dermal fibroblasts) and PXEF(PXE fibroblast) under various conditions. Although the manuscript is well written few queries need to be addressed before suitable publication.

Major comments

1)      Figure 1- The authors here have shown upregulated matrix metalloproteinases expression upon ABCC6- deficiency by RNA-Sequencing. It is highly recommended to deposit the raw sequencing data to an open-source repository (e.g. Gene Expression Omnibus – NCBI). Also, in the methodology please provide a detailed explanation for PXEF test samples and NHDF controls used and if then how many replicates (biological) were used to perform RNA-Seq. Discuss in detail how the RNA-Seq experiment was performed, and the analysis was done.

Also, give a graphical representation of the differentially expressed genes (DEG’s) or volcano plots for this experiment. (This could be added as supplementary data to the manuscript).

2)      Figure 2A. The authors should validate the downregulated MMPs(hMMP24 and hMMP15) alongside upregulated MMPS.

3)      Figure 2B. The authors have shown murine fibroblast(ABCC6 wt vs mutant) does not show a significant difference in MMPs expression although mutant ABCC6 shows ~20 fold more mMmp10 expression over wild type. Overall authors should comment on the choice of performing this experiment with the murine model.

4)      Figure 2 D: The image quality provided in the manuscript is very low. The magnification for NHDF and PXEF in both D, and E looks different from each other. Please make sure the magnification is justified for control vs test samples used.

 5)      Figure 3: The cell density observed for control and +10uM Marimastat for NHDF and PXEF are different. Please explain this finding.

6)      Figure 4(A, B, C, D)- The legends provided for this figure is not clear and easy to understand. The comparison between the groups shown in the figure is very confusing. Please simply the group comparisons made and justify accordingly.

Minor comments

1) Table 3. The gene names are written as human(h). Please correct it to murine(m)

2) Although the manuscript is well written, it is highly recommended to proofread the manuscript by a someone who is native English speaker. 

Reviewer 2 Report

The paper by Plümers et al provides new insights on the role of MMPs in the ethiopathogenesis of PXE by inducing calcification. The study is interesting, and used murine ABCC6-/- fibroblasts as well as human dermal fibroblasts from PXE patients.

Comments from the reviewer:

1)      The authors should state the number of mice used/group. This is stated in some figure legends but should also be mentioned in material and methods. Furthermore, there is no information about how many males and females were considered.

2)      The study showed an overexpression of MMP12 near the nuclei by immunofluorescence. The authors state that MMP12 is predominantly secreted, rather than accumulated inside the cell. To confirm this statement, the study would benefit from an ELISA assay for his MMP in the cell culture supernatant, identical to the one performed for MMP3.

3)      Data in figure 3 reveal different cell shape phenotypes. This might just reflect the reduction in cell growth observed in these cultures. A graph showing the proliferative rates of the distinct cultures should be included in this experiment.

4)      It is interesting that human MMPs mRNA were significantly induced in PXEF, but this was not significant in murine cells, although an increase was also observed. This finding should be mentioned in Discussion.

5)      Further, the authors report that human and murine MMPs are closely related. Nevertheless, they identified some differences in MMPs between the two species. This should further be addressed in Discussion. Why were hMMP8, 12, 10, 3, 27, 1 and 2 upregulated in PXEF, and mMmp10, 1a, 13, 1b and 3 in mice EPXF? Do these MMPs present overlapping roles?

6)      Please correct sentence in line 377: “By contrast, application of only Marimastat resulted …”. It is not clear.

7)      The p value for ** is lacking in Figure 4D

8)      There are several misspellings and grammatical mistakes throughout the manuscript (e.g. “Exemplary” used instead of “representative”, “In the following…”, “decryption”, etc)

9)      Please clarify sentence in line 480: “As the disease is characterized by…”

10)   Sentence in line 514 is not completed.

Reviewer 3 Report

The manuscript by Plümers et al, described RNA seq data obtained from skin fibroblasts derived from PXE patients and controls in addition to Abcc6-/- mice. Additonal in vitro work was done to demonstrate the role of MMPs in the calcification phenotype of PXE.

This manuscript is not particularly well-written and suffers from grossly over interpretion of limited results. 

Specific comments.

-       Some improvement of the English is need.

-       The references of the first 2 paragraphs are all wrong.

-       The use of elevated passage numbers (8-11) for primary fibroblasts is questionable

-       Abcc6-/- mice do not present skin calcification (except in vibrissae) and certainly not in the tail. Arguably, vibrissae has no equivalent structure in human yet it is used a disease progression marker in Abcc6 KO. However, the rational for deriving fibroblasts from the tail skin is not explained and is an very odd choice as there is equivalent tissue in human, Overall, mouse skin is quite different from humans, notably with respect to important ECM component such as elastic fibers which are very limited in mice.

-       Why using high passage number primary cells when several of the experiments, transcriptome, immunostaining could have been done in situ in biopsy samples, this is especially true with animal samples which are more accessible.  

-       The rationale for inducing calcification with elevated calcium is surprising as most in vitro calcification models used increased phosphate. This is even more surprising as there is no change to calcium levels in PXE whereas the Pi/PPi ratio is altered. 

-       Fig 2, 4, 5. Are n-numbers replicates of the same cells or do they correspond to different patients/mice ?

-       Fig. 2D and E are problematic. D: sizes of apparent nuclei (DAPI) are different between normal and PXE and the immunostaining quality is sub-par. There is no negative controls. E: calling this staining as endoplasmic reticulum is speculative at best without actually demonstration. And the localization may infact indicate a problem with the antiboby. Verifying the results with another antibody is indicated and so are negative controls. Why wasn’t there mouse fibroblast immunostaining too?

-       Fig 3 is not informative especially at the magnification chosen and the associated observations (morphology changes) are not quantified so this amounts to circumstantial evidence. The significance of the morphology changes is not discussed. 

-       Furthermore, the observed morphology changes could be related to passage numbers as elevated passage numbers and/or Abcc6 knockout lead senescence and morphology changes Cf. Miglionico et al (Ref 46).

-       The lack of PPi measurements in the culture supernatant is quite surprising especially with ENPP1gene expression looked at.

-       The authors should perhaps have treated the fibroblast cultures with PPi and/or etidonate instead of Marimastat – this would have been far more informative.

-       Discussion: why is chromosomal clustering of MMP genes remarkable? What is the relevance to PXE?

-       There is no discussion on M vs F differences

-       The mouse data does not fully support the human results

-       The section on Inflammatory cytokines as regulator of gene expression is highly speculative in the absence of any direct evidence.

Overall, the paper is a small incremental step as MMP differences in PXE have already been shown. In vivo data would be far more relevant to the pathobiology of PXE. The paper fails to establish a credible connection between enhanced MMP expression and calcification and the discussion stretches thin and mixed results. 

Reviewer 4 Report

This manuscript characterized human and murine fibroblasts with ABCC6-mutation which results in pseudoxanthoma elasticum (PXE). The authors examined elevated expression profiles of matrix metalloproteinases (MMPs) and discovered that MMP inhibitor Marimastat did not abolish excessive calcification but reinforced it in their in vitro models. In general, the problem is well motivated, and there is clearly a strong need for mechanistic studies. Key issue to focus on is consistent experimental validations across various MMP groups.

Major comments: 

  1. The expression levels of different MMPs are different and so are their responses to Marimastat treated calcification conditions. How is the Marimastat concentration determined in this paper? If MMP3 and 12 may different roles in PXE, would it make more sense to have specific inhibition/down regulation experiments to verify their roles, instead of group them together?
  2. In figure 2, If the mRNA levels were highly elevated for MMP1, 3, 10, and 12, it would seem a bit cherry picking if only MMP3 ELISA levels and MMP 12 immunofluorescence results were shown as functional confirmation. To fully characterize the MMP expression profile, it will make the arguments stronger if the authors can either add MMP3 immunofluorescence staining similarly to MMP12 or show the ELISA profile for MMP12. Similarly, in figure 4, it is incomplete if MMP1, 3, and 12 mRNA levels were compared without MMP10. Also, it is confusing why only MMP3 protein levels were presented in the figure while other MMP types were not compared. 
  3. The surprising effects of MMP inhibition led to increased calcification is quite striking. Since the main function of MMP is related to ECM degradation, perhaps the authors need to evaluate the cell behaviors not just on tissue culture plastics, but also coated with ECM proteins. In figure 6, it is also confusing why despite higher calcification status, the overall gene expression levels of calcification markers were lower in PXE fibroblasts compared to normal fibroblasts. The authors need to expand the discussion section to explain this discrepancy.

Minor comments:

  1. In method 2.6, the immunofluorescence staining is for MMP12, not MMP2 in the section title. 
  2. In figure 2B, the MMP3 and MMP10 changes are quite dramatic but they were labeled as “ns”. Why is a 9.5-fold change not significant from statistical analysis? 
  3. The statistical labeling for figure 4 is confusing. It appears that there may be some misalignment of line and stars and the authors should double check on it. 

Round 2

Reviewer 2 Report

The authors addressed carefully all the comments of the reviewer. The manuscript can be accepted.